# I♥LA: Compilable Markdown for Linear Algebra

Yong Li
George Mason Univ.

Shoaib Kamil
Adobe Research

Alec Jacobson
Univ. of Toronto
Adobe Research

Yotam Gingold
George Mason Univ.

I♥LA is a compilable Markdown-like programming language for linear algebra. It can generate working code in C++ and Python (and more to come). The same I♥LA code can also generate LaTeX, in turn rendered as beautifully typeset math. The I♥LA code creates a publishable artifact and reference implementation. We believe I♥LA can improve expositional clarity, code reproducibility & interoperability, and scientific education. We focus our initial efforts on mathematical expressions found in the wild within the Computer Graphics community, but plan to extend our work to the larger Machine Learning and Computer Science community.

In our proposed Rethinking ML Papers exhibit, we will demonstrate:

- results of our large-scale study, and

- an interactive demo application.

We hope that this exhibit will encourage new users to try I♥LA, and help advance code replicability and communication in communities that heavily rely on math.

## 1 Overview of I♥LA

We designed the language to be as close as possible to the "natural" math syntax used in Computer Graphics and other communities, while ensuring that I♥LA programs have an unambiguous interpretation. To do this, we tabulated all 1,994 numbered equations in the ACM SIGGRAPH 2019 technical papers proceedings. A key decision was to embrace the use of Unicode characters, such as $\Sigma$ and $\pi$, as a fundamental part of the language. We also eschew a multiplication operator in favor of juxtaposition, as commonly done in written math.

Our compiler, written in Python, transforms I♥LA programs into a typed, high-level intermediate form, checking to ensure required properties (such as matching dimensions), and from there into semantically-correct C++ programs that utilize the Eigen math library; Python programs that use NumPy/SciPy; and LaTeX output for inclusion in papers or on the web (via MathJax).

We currently have two user interfaces: a desktop client and an in-browser client.

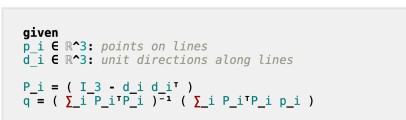

```
given
p_i ∈ ℝ^3: points on lines
d_i ∈ ℝ^3: unit directions along lines

P_i = ( I_3 - d_i d_iᵀ )
q = ( ∑_i P_iᵀP_i )⁻¹ ( ∑_i P_iᵀP_i p_i )
```

Figure 1: For example, this I♥LA code compiles to a function that computes the closest point $q$ to a set of 3D lines.

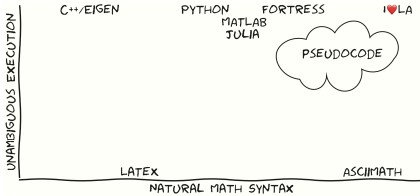

Figure 2: I♥LA combines natural syntax with unambiguous execution.

## 2 Large-Scale Study Gallery

In our accompanying technical paper Li et al. (2021), we describe a large-scale study where we apply I♥LA to write equations found in Computer Graphics papers and applications. Our exhibit will include a web tour of this study, showing the concise I♥LA input and the semantically-correct outputs for each example.

## 3  Interactive Demo Application

As part of our exhibit we will provide an in-browser I❤LA compiler that attendees can use to try the language. Please see accompanying website at https://cragl.cs.gmu.edu/iheartla/rethinkingpapers/ for an explanatory video, the aforementioned gallery, and an in-browser demo of the I❤LA prototype integrated development environment (IDE).

## 4  Accessibility Statement

We commit to providing an exhibit that is accessible to the widest possible audience, regardless of technology or ability. We are actively working to increase the accessibility and usability of our IDE and example gallery website. Our website and in-browser demo endeavour to conform to level Double-A of the World Wide Web Consortium (W3C) Web Content Accessibility Guidelines 2.1. These guidelines explain how to make web content more accessible for people with disabilities. Conformance with these guidelines will help make the web more user friendly for all people. Our exhibit is built using code compliant with W3C standards for HTML and CSS. The site displays correctly in current browsers and using standards compliant HTML/CSS code means any future browsers will also display it correctly. While we strive to adhere to the accepted guidelines and standards for accessibility and usability, it is not always possible to do so in all areas of the website. For example, rendered LaTeX is challenging for screen readers. Our MathJax output serves as an alternate via its Accessibility extensions. We are continually seeking out solutions that will improve accessibility. Should anyone experience any difficulty in accessing our exhibit, please contact us at yli69@gmu.edu.

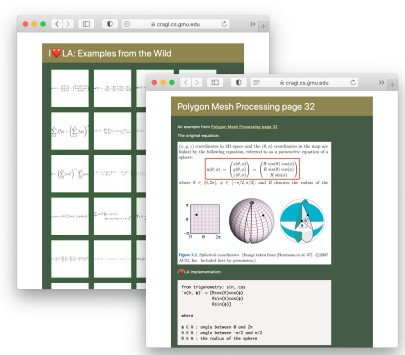

Figure 3: Our gallery contains I❤LA reimplementation of 26 complicated equations found in the computer graphics literature and 10 examples of I❤LA code integrated into existing code bases.

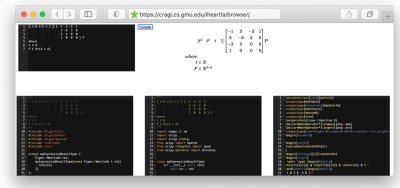

Figure 4: Our exhibit will feature an in-browser, interactive demo.

## 5  Discussion & Future Directions

We imagine I❤LA leading to increased interoperability, as technical innovations and ideas can be expressed once and then compiled into various programming and paper ecosystems. This can increase access to computer science through a language closer to natural mathematics. We plan to explore integrations with distil.pub-type interactive documents and Jupyter notebooks. Like a compilable Markdown, I❤LA Jupyter notebook cells can both display as compiled LaTeX and run as code in the notebook's kernel.

We plan to create a converter from handwritten equations to I❤LA, improve the embeddability of I❤LA within existing languages, and explore applications of the IR. We also plan to expand the syntax to formula used in other disciplines, such as tensor math in ML.

## References

Yong Li, Shoaib Kamil, Alec Jacobson, and Yotam Gingold. I❤LA: Compilable markdown for linear algebra. Technical report, 2021. URL https://cragl.cs.gmu.edu/iheartla/rethinkingpapers/preprint.pdf.

