# OpenReview forum: "I❤LA: Compilable Markdown for Linear Algebra"
_ICLR.cc/2021/Workshop/Rethinking_ML_Papers/Exhibit_and_Workflow — Rethinking ML Papers - ICLR 2021 workshop Oral_

### Official Review · Reviewer_rxv9 · 2021-03-29
**Domain-specific language for linear algebra**

**Accessibility:**

Score of 5 (Exceptional): Submission identifies and articulates accessibility matters, provides justifications for the proposed paradigm, and declares the limitations.

**Litreview:**

Score of 3 (Neutral): The submission acknowledges previous work, but does not necessarily explain how the submission differentiates itself (i.e we want to avoid the “deluge of citation” strategy, leaving the reviewer to click through references and figure this part out for themselves).

**Problemstatement:**

Score of 5 (Exceptional): The submission states a well-known problem relevant to the workshop, and sets what could be a new standard in the field when it comes to addressing it.

**Relevance:**

Score of 4 (Strong): The submission directly addresses a theme of the workshop, and does so in a very professional manner.

**Results:**

Score of 5 (Exceptional): Submission has an excellent design and all criteria are addressed. Conclusions, practical/theoretical implications are well articulated.

**Reviewerconfidence:**

I would rank my confidence in this review as a 4 given I have experience contributing to open-source DSLs and have colleagues who maintain large projects, but have not created nor evaluated one myself.

**Reviewtext:**

I❤ LA is a compilable Markdown-like domain specific language (DSL) for linear algebra equations. Although all domain-specific languages require learning new syntax, the capability to write pseudo-code text and have it compiled into C++, Python, and LaTeX code at once is very compelling, and I could envision many different applications for this tool. I appreciate the web demonstration, including the video, examples from the wild, and in-browser editor to get one’s hands on the system. To further improve the the language, including evaluating the syntax, polishing the editor, and supporting the DSL moving forward, I would look at other popular DSLs to see their features/demonstrations/papers (e.g., https://vega.github.io/editor/, https://idyll-lang.org/editor).

This project is a great effort, and as a consequence could also help computer science and applied math educators transition from linear algebra theory to working examples. This could also help someone who does not know how to program move theoretical ideas to code.

My only suggestion to improve the written artifact is to include a short discussion around how this compiler works, what it can support, its current limitations, and the future plans for the DSL.


**Score:**

Strong accept: The reviewer has a strong enthusiasm to apply the proposed framework in their work.

---

### Official Review · Reviewer_eU22 · 2021-03-30
**Promising compilable markdown language.**

**Accessibility:**

Score of 4 (Strong): Submission states accessibility concerns and provides solutions within the proposed framework. However, it does not declare the limitations and exceptions.

**Litreview:**

Score of 2 (Needs Improvement): The submission leaves out prominent examples of previous work in the area.

**Problemstatement:**

Score of 2 (Needs Improvement): The submission clearly has potential or credibility, but still fails to state the problem addressed clearly.

**Relevance:**

Score of 4 (Strong): The submission directly addresses a theme of the workshop, and does so in a very professional manner.

**Results:**

Score of 4 (Strong): Submission is very well structured and follows all the criteria (i.e. clarity, novelty, interactivity, and coherency). However, practical significance/theoretical implications are not discussed.

**Reviewerconfidence:**

3: I can understand the essence of the interactive tools like the one presented on the paper. But I haven't worked firsthand on the "development" of these tools and am unaware of the literature's current state.

**Reviewtext:**

I❤LA generates cross-platform codes using the commonly used math style language. Different platform supported by the current version includes popular languages in ML development including Python and C++, and widely used LaTeX formatting.

Strengths:
- Ability to generate cross-platform codes using the commonly used math style language. This ability seems scalable with extension to other languages shortly.
- As shown in the large-scale study, the platform supports an extensive range of mathematical equations found in Computer Graphics papers and applications and thus, aptly supports their plan to extend to the larger ML or CS community.
- Clear and understandable demo.
- Clear plans to make the current platform accessible.

Weakness:
- Lack of discussion around related works and frameworks that are working along with the same objective.
- One of the essential steps within accessibility is to allow the users to send feedback, which currently seems to be missing in their framework. However, it's worth noting that the authors stated, "we will provide a way for them to contact us".

Minor suggestions:
- Although Linear Algebra is one of the important mathematical sub-disciplines within ML or CS, there are other disciplines (e.g., differential geometry, etc.) that are continuously used by the ML community to advance its breath. I suggest authors/developers have a broader name for their framework.

**Score:**

Accept: The reviewer believes the submission provides a novel and reliable scheme to improve science communication but needs improvement.

---

### Meta-Review · Area_Chair_AUjs · 2021-03-31

**Recommendation:** Accept
**Confidence:** 4

**Metareview:**

Exciting and technically sound proposal of markdown programming language for linear algebra. The proposed framework is very relevant to many applications and includes results of our large-scale study and an interactive demo.

Reviewers have a consensus that this would be a great addition to the workshop. I recommend acceptance.
Please incorporate reviewer feedback in camera-ready such as more thorough discussion on related work, improve the written artifact is to include a short discussion around how this compiler works, what it can support, its current limitations, and the future plans for the DSL.

---

### Decision · Program_Chairs · 2021-04-01

Accept (Oral)